# Genetic algorithm with a new round-robin based tournament selection: Statistical properties analysis

**Abid Hussain** [1]*, **Salma Riaz**[2], **Muhammad Sohail Amjad**[3], **Ehtasham ul Haq**[4]

1 Department of Statistics, Govt. College Khayaban-e-Sir Syed, Rawalpindi, Pakistan, 2 Department of Applied Mathematics & Statistics, Institute of Space Technology, Islamabad, Pakistan, 3 School of Applied Sciences & Humanities, National University of Technology, Islamabad, Pakistan, 4 Department of Mathematics and Statistics, International Islamic University, Islamabad, Pakistan

* abid0100@gmail.com

## Abstract

A round-robin tournament is a contest where each and every player plays with all the other players. In this study, we propose a round-robin based tournament selection operator for the genetic algorithms (GAs). At first, we divide the whole population into two equal and disjoint groups, then each individual of a group competes with all the individuals of other group. Statistical experimental results reveal that the devised selection operator has a relatively better selection pressure along with a minimal loss of population diversity. For the consisting of assigned probability distribution with sampling algorithms, we employ the Pearson's chi-square and the empirical distribution function as goodness of fit tests for the analysis of statistical properties analysis. At the cost of a nominal increase of the complexity as compared to conventional selection approaches, it has improved the sampling accuracy. Finally, for the global performance, we considered the traveling salesman problem to measure the efficiency of the newly developed selection scheme with respect to other competing selection operators and observed an improved performance.

## 1 Introduction

Genetic algorithms (GAs) are stochastic approaches for optimization, based on natural mechanisms of genetics. These algorithms refer to the natural selection process where the most fitted individuals for reproduction. Generally, five stages are considered in GA: initial population, fitness function, selection, crossover and mutation operators. If parents have good fitness, their offspring will be better than them and have a better chance to survive. The process continues and eventually a generation is found with the most qualified individuals.

The development of GAs originates from the influential work of Holland [1]. Many scholars have acknowledged GA as it is considered a key member of optimization related research. The global environment, robustness actions and reliability are the main reasons for its popularity. For example, Song et al. [2] employed GAs to achieve optimal satellite selection for global positioning system (GPS) use. Ha et al. [3] proposed hybrid-GA for traveling salesman problem

**Data Availability Statement:** The data used to support the findings of this manuscript are taken from the website of TSPLIB (http://comopt.ifi.uni-heidelberg.de/software/TSPLIB95/).

**Funding:** The author(s) received no specific funding for this work.

**Competing interests:** The authors have declared that no competing interests exist.

with drone to deliver parcels to customers. Recently, Wang et al. [4] explored GAs in optimizing a credit portfolio while minimizing the default risk under the constraint of a target expected premium. Other than these, many applications of GAs can be perceived in the multidisciplinary research literature, such as, lung cancer prognosis [5], for the fuzzy shortest path problem in a fuzzy network [6], for Designing envelope configurations of building with the low construction cost and low energy consumption [7], the detection of software vulnerabilities [8], for data mining tasks [9], for nutritional Anemia disease classification [10] and large-scale and dynamic social networks [11]. Someone can consult to Katoch et al. [12] for a detailed overview on GAs with several applications.

In all domains of life, GAs are found to be effective but still there is the issue of premature convergence in pursuit for optimal solutions, see, for example, Hussain and Muhammad [13]. The complications of premature convergence are entrenched in the philosophical orientation of GAs as shortened by Julstrom [14]. Maintaining good diversity in the population is required to GA success. Otherwise, this leads to stuck off on local optima, which is an undesirable situation in GA, called premature convergence. The optimization literature acknowledging the diversity of population is a vital factor in search of a global optimal solution. This is evident by the discussion on the relevance of premature convergence and population diversity, see, for example, Hussain andMuhammad [13, 15]. So it is clear from these studies that the performance of GA is mostly affected by the choice of selection operator. The selection operator is the most crucial research area in the body of contributions associated with GAs.

Due to the importance of the selection phase in GAs, the current research contributes to the literature by introducing a novel selection operator, namely the round-robin based tournament selection (RRTS). The main focus of this research is on facilitating the convergence process by maintaining a desirable level of population diversity. This objective is accomplished by achieving a tradeoff between exploration and exploitation. The fitness rank of participants in concordance with the normality of generations is used to aid the selection process and the encouraging results of this delicate selection scheme are documented in this article.

The remainder part of this paper is organized as follows. In Section 2, the selection operator as a two-stage procedure, with a detailed review, has been discussed. In Section 3, we propose a new selection operator with its theoretical and mathematical foundations. Further, several stochastic properties of the newly proposed operator are reviewed in Section 4. Inspired by the stochastic features, Section 5 delineates the applicability of the proposed methodology in solving one of the practical problems, i.e. the traveling salesman problem (TSP). Lastly, Section 6 summarizes the study along with a brief discussion of future perspective research.

## 2 Selection procedure

The selection process in GA can be split into two stages. In the first stage, a selection probability is assigned to each and every individual based on fitness values. These probabilities are denoted as: $P = (p_1, \ldots, p_K)$, where $p_i \in [0, 1]$ and $\sum_{i=1}^{K} p_i = 1$ with $K$ is the population size. To investigation about selection probabilities, someone can consult to Hussain and Muhammad [13] and Jul-strom [14]. The second stage is the sampling process, which selects the most fitted parents (based on Darwin's "survival-of-the-fittest" criterion) from the current population for mating process. A thoroughly discussion about sampling algorithms has been provided in Section 4. This study, additionally, has a significant effect on the GA's selection methods. In this perspective, a new operator for the selection is projected that is expected to reinforce the typical character of the population and offers an improved tradeoff between exploitation and exploration.

## 2.1 Assignment of probability

The first and the most popular selection procedure known as fitness proportional selection (FPS), was proposed by Holland [1]. In this selection procedure, the selection probability of $i^{th}$ individual, say $p_i$, is directly proportional to its fitness. The theme of this method depends upon the understanding that fitter individual ought to have a high probability of selection, whereas each individual to become the member of the parent population using the following formula:

$$p_i = \frac{f_i}{\sum_{j=1}^{K} f_j}; \qquad i \in \{1, 2, ..., K\}, \tag{1}$$

where, $f_i$ denotes the fitness status of $i^{th}$ individual.

The operational directives of FPS are similar to the probability proportional to size (PPS) sampling using with replacement approach. Throughout the entire selection process, there will be no alternation needed in size and possibilities. This method is easy to enforce and offers probabilities to all individuals according to their fitness values, but the scaling problem is its main drawback, see, for instance, Grefenstette [16].

The linear rank selection (LRS) was introduced by Baker [17], is catered as the remedy of premature convergence attributed with FPS. This method provides a relatively better opportunity to pick out weaker individuals and therefore offered a smoother selection function. In the LRS procedure, the $i^{th}$ individual is assigned a selection probability using the following formula:

$$p_i = \frac{1}{K} \left( \vartheta^- + (\vartheta^+ - \vartheta^-) \frac{i-1}{K-1} \right); \qquad i \in \{1, 2, ..., K\}, \tag{2}$$

where, $i$ is the rank of the individual based on fitness status and $\vartheta^-$ and $\vartheta^+$ are the parameters for the selection probabilities of worst and best individuals based on their ranks, respectively. The two constraints which are associated with this scheme as: $\vartheta^+ + \vartheta^- = 2$ and $\vartheta^- \geq 0$. As a result, even if individuals differ notably in fitness status, the ranks remain work in uniform pattern, unable to reflect the difference with desirable intensity and so naturally compromise relevant data. Along with numerous applications of LRS, as an example, see, Sharma and Mehta [18], but on the other hand, it has a drawback of slower convergence of the algorithm. This is because of its internal methodology based on ranks instead of fitness values directly for selection, see, Aibinu et al. [19] and Hussain and Muhammad [13]. This issue becomes more serious in the case of a larger population where ranks are thought about as a realization from the uniform distribution. To resolve the difficulty of LRS, Michalewicz [20] designed an alternative rank based selection operator, called exponential rank selection (ERS). To tell apart from LRS, Michalewicz [20] prompt that the selection probabilities increase exponentially from worst individual to best one. The selection venue of individual medaled with $i^{th}$ rank, is mathematically written as:

$$p_i = \frac{v^{K-i}(1-v)}{1-v^K}; \qquad i \in \{1, 2, ..., K\}, \tag{3}$$

where, $0 < v < 1$, $v$ is a fixed ratio defined as weights of individuals based on their fitness ranks and maximum gain values of $v$ closer to unity (i.e. $v \to 1$) is recommended by Michalewicz [20]. ERS as a popular selection method is evidenced by various applications, see, for example, Schell and Wegenkittl [21] and Lee et al. [22].

Another selection procedure, which is based on a real phenomenon of the tournament, known as binary tournament selection (BTS), was introduced by Back [23]. Using BTS, two

competitors are randomly chosen, then a winner will be selected for mating process. In this case, the chance of choosing a good parent is very high, but if both of the selected parents are of low quality, the low quality parents will be selected. Spotting the significance of population diversity, Back [23] insisted lower tournament size because pair wise comparisons remain the most common theme in tournament selection schemes. The selection probability of $i^{th}$ ordered individual is given as:

$$p_i = \frac{1}{K^r}((i)^r - (i-1)^r); \qquad i \in \{1, 2, ..., K\}, \tag{4}$$

where, $r$ represents the array of tournament size.

Julstrom [14] employed a probability-based threshold level to select the winner of the tournament called probabilistic 2-tournament selection (PTS). In this scheme, the competition winner will be survive with a probability $0.5 < q < 1$, where the loser will get another chance of competing, with probability $1 - q$. For the PTS method, the $i^{th}$ ordered individual is assigned the selection probability by the following rule:

$$p_i = \frac{2(i-1)}{K(K-1)}q + \frac{2(K-i)}{K(K-1)}(1-q); \qquad i \in \{1, 2, ..., K\}. \tag{5}$$

This selection procedure has great applicability, see, for instance, Schell and Wegenkittl [21] and Lee et al. [22].

In recent past, Hussain and Muhammad [13] suggested a new split-based rank selection (SRS) to tradeoff between exploitation and exploration. In their scheme, all individuals are ranked according to their fitness values and assigned probabilities by using the following formula:

$$p_i = \begin{cases} \lambda^- \left( \frac{8i}{K(K+2)} \right); & i \leq \frac{K}{2} \\ \lambda^+ \left( \frac{8i}{K(3K+2)} \right); & i > \frac{K}{2}, \end{cases} \tag{6}$$

where, $\lambda^+ + \lambda^- = 1$ with $\lambda^- \geq 0$ must be satisfied. The selection pressure can be restrained by varying $\lambda^+$, the tuning parameter, in the selection phase.

The most significant feature of the selection operator is the selection pressure because it is adjacent with a suitable balance between exploration and exploitation. Eiben et al. [24] described a scenario, where a relatively lower selection pressure is required at the initial stage for diversity in the whole sampled population and enhance at the last stage to assist the convergence of algorithm. To tradeoff between the two extremes, an adjustable selection pressure should be required, see, for example, Pham and Castellani [25].

This article proposes a new selection approach, which removes the weakness related to fitness based (i.e. FPS), rank based (i.e. LRS and ERS) and tournament based (i.e. BTS and PTS) approaches. It is predicted on the basis of a tournament scheme, whereas we split all the individuals into two equal and mutually exclusive groups and assign them probabilities for selection according to their ranks. The details that how an individual is competing with other group's members to be survived as parent for mating process are provided in the adjacent section.

## 3 The proposed selection operator

### 3.1 Motivation

There are many selection mechanisms that have been proposed in the literature. As LRS emphasized maintaining higher levels of population diversity at the cost of selection pressure and results in slowest convergence of GAs. On the other hand, the FPS method has high selection pressure with sacrificing the diversity and as a result remains the prime candidate of suffering from premature convergence. In this section, a new operator capable of achieving more balance between exploration and exploitation is proposed, which provides sufficient selection pressure throughout the selection process.

### 3.2 Round-robin based tournament selection

An alternative selection scheme (round-robin based tournament selection (RRTS)) is proposed to maintain a precise balance between exploration and exploitation. In this approach, an adequate selection pressure with elimination of the fitness scaling problem is provided. Consider the following steps for the proposed selection procedure:

1. In the RRTS method, all individuals are ranked according to their fitness measures and acquire a distinct rank even though they have equivalent fitness values.

2. The individuals are divided into two equal and disjoint groups, e.g. $A$ and its complement $A^c$. The population can be inserted in these groups in multiple ways, such as: randomly, first half is in one group and rest is in other (best-worst), the odd individuals are in one group and even in the other (even-odd), up to 25% and 51% to 75% in one group and rest in other etc.

3. Now, one individual, i.e. $i$, is chosen at random from a group with a surviving process probability $\theta$, also comparing the combined effect of all the other group members with probability $(1 - \theta)$. Thus, the selection probability to select an individual as a parent is determined by the subsequent rule:

$$p_i = \frac{2(i-1)}{K(K-1)}\theta + \frac{4(1-\theta)}{K^2(K-1)}\sum_{j \in A^c}(K-j), \qquad (7)$$

where if $i$ belongs to one group then $j$ belongs to other group and $K$ is the population size.

Table 1 presents some rules to assign probabilities to all the individuals for $K = 10$ and $\theta = 0.5$. There are two tuning parameters to maintain a tradeoff between diversity and selection pressure in our proposed method, i.e. the value of $\theta$ and group segmentation.

## 4 The sampling algorithms

The first stage in the selection phase is to assign probabilities to all competing individuals, whereas in the second stage, a sampling algorithm is requisite to fill the mating pool for parents, whereas this process reflects the selection probabilities, such that the expected and observed number of individuals are equal. In this study, two popular sampling methods, roulette wheel sampling (RWS) and stochastic universal samplings (SUS) are used for testing.

### 4.1 Roulette wheel sampling

The roulette wheel sampling (RWS) was introduced by Holland [1] and it is still one of the most popular sampling methods for GA. In the RWS procedure, each possible solution is

**Table 1. Selection probabilities using various criteria in proposed method.**

| Rank(i) | Randomly | | Even-odd | | Best-worst | |
|---|---|---|---|---|---|---|
| | Group | $p_i$ | Group | $p_i$ | Group | $p_i$ |
| 1 | $A$ | 0.0489 | $A$ | 0.0444 | $A$ | 0.0222 |
| 2 | $A^c$ | 0.0622 | $A^c$ | 0.0667 | $A$ | 0.0333 |
| 3 | $A^c$ | 0.0733 | $A$ | 0.0667 | $A$ | 0.0444 |
| 4 | $A$ | 0.0822 | $A^c$ | 0.0889 | $A$ | 0.0556 |
| 5 | $A^c$ | 0.0956 | $A$ | 0.0889 | $A$ | 0.0667 |
| 6 | $A$ | 0.1044 | $A^c$ | 0.1111 | $A^c$ | 0.1333 |
| 7 | $A$ | 0.1156 | $A$ | 0.1111 | $A^c$ | 0.1444 |
| 8 | $A^c$ | 0.1289 | $A^c$ | 0.1333 | $A^c$ | 0.1556 |
| 9 | $A$ | 0.1378 | $A$ | 0.1333 | $A^c$ | 0.1667 |
| 10 | $A^c$ | 0.1511 | $A^c$ | 0.1556 | $A^c$ | 0.1778 |

appointed as a slice with respect to a portion, which is assigned by a desired selection probability method. By using a single marker at the border of the roulette wheel and the roulette wheel is spun $K$ times to successively select individuals. This sampling method is very simple and easier to implement with a high probability for the better choice of chromosomes. Clearly, the vector $(o_1, o_2, \ldots, o_K)$ follows a multinomial distribution with parameters $K$ and $P$, where $P = (p_1, \ldots, p_K)$. The mean and variance of this distribution are given below:

$$E[(o_1, o_2, ..., o_K)] = (e_1, e_2, ..., e_K),$$

$$V[o_i] = Kp_i(1 - p_i).$$

## 4.2 Stochastic universal sampling

The mechanism of stochastic universal samplings (SUS) was introduced by Baker [17] and is quite similar to RWS. The only difference between RWS and SUS is the number of markers: one marker in RWS and $K$ (population size) number of markers in SUS. In this method, $K$ markers spaced evenly are used at the border of the roulette wheel. The slices of wheel are consistent as in RWS. In the SUS method, the roulette wheel is spun one time only and select all individuals, which are pointed by the $K$ markers and enclosed in the mating pool as parents. Therefore, all the parents are chosen in just one cycle of the wheel and this method promotes the better individuals for selecting at least once.

The computational complexity of SUS (i.e. $O(N)$) is comparatively lower than the complexity of RWS (i.e. $O(N^2)$), to identify the selected candidates as parents, only one pass over the population is needed. However, their expectations close with each other but the variabilities of the most fitted individuals are significantly least than RWS.

A detailed comparison between two sampling methods, i.e. RWS and SUS can be founded in the central moments of the distributions of the vectors $(o_1, o_2, \ldots, o_K)$. The absolute difference between an individual's observed and its expected values is defined as bias, i.e. $|o_i - e_i|$. In sampling, each individual might be provided a certain number of copies that are placed into the mating pool. The possible range of the number of copies is called "spread". The SUS provides the insurance of minimum spread and almost zero bias.

### 4.3 The chi-square test as a goodness-of-fit measure

For empirical analysis, the chi-square test is a measure to ascertain the accuracy of sampling algorithms, i.e. RWS and SUS, and will compare with the probability distribution of the selection operators. As a tool for measuring the expected accuracy, the $\chi^2$ test was first introduced by Schell and Wegenkittl [21].

Let we consider, $\xi_j = \sum_{i \in C_j} e_i$ is an overall expectation, whereas $O_j = \sum_{i \in C_j} o_i$ be the observed (actual) copies of individuals in mating pool after the sampling procedure. The two disjoint classes are: $\{C_1, C_2, \ldots, C_c\}$, $C_j \subset \{1, 2, \ldots, K\}$ and $\cup_{j=1}^{c} C_j = \{1, 2, \ldots, K\}$. For expected behavior, the $\xi_j$ be of the enjoin $K/c$ members with $1 \leq j \leq c$, to regulate that each class maintains the same number of individuals (on average). To desired stochastic accuracy, at least 10 individuals should be in each class. The chi-square test is defined as:

$$\chi := \sum_{j=i}^{c} \frac{(\xi_j - O_j)^2}{\xi_j}. \tag{8}$$

In the RWS algorithm, the $\xi_j \geq 10$ as it minimizes the differences between expected and observed frequencies. On the other hand, for SUS, we expect that $\chi \approx 0$. In Table 2, the probability distributions of all competing selection operators with the corresponding overall expected individuals (i.e. close to 300/10) are presented. $\chi^{S, R}$ is the measure of chi-square for operator $S$ that assigns the probabilities to individuals and for sampling algorithm $R$.

The basic purpose of this test is to describe the sample mean and sample variance. The initial sampled population is to be considered as randomly. The probability distribution $S$ is used

Table 2. The overall expected counts, $\xi_j$, with respect to their classes, $C_j(j = 1, 2, \ldots, 10)$.

| j | LRS | | ERS | | BTS | |
|---|---|---|---|---|---|---|
| | $C_j$ | $\xi_j$ | $C_j$ | $\xi_j$ | $C_j$ | $\xi_j$ |
| 1 | 1–33 | 30.05 | 1–108 | 30.33 | 1–95 | 30.08 |
| 2 | 34–65 | 29.84 | 109–158 | 29.91 | 96–134 | 29.77 |
| 3 | 66–96 | 29.56 | 159–192 | 30.84 | 135–164 | 29.80 |
| 4 | 97–127 | 30.20 | 193–217 | 30.43 | 165–190 | 30.68 |
| 5 | 128–157 | 29.84 | 218–237 | 30.50 | 191–213 | 30.90 |
| 6 | 158–187 | 30.44 | 238–253 | 29.22 | 214–233 | 29.73 |
| 7 | 188–216 | 30.00 | 254–267 | 29.72 | 234–252 | 30.72 |
| 8 | 217–244 | 29.50 | 268–279 | 29.02 | 253–269 | 29.52 |
| 9 | 245–272 | 30.02 | 280–290 | 29.86 | 270–285 | 29.55 |
| 10 | 273–300 | 30.54 | 291–300 | 30.16 | 286–300 | 29.25 |
| j | PTS | | SRS | | RRTS | |
| | $C_j$ | $\xi_j$ | $C_j$ | $\xi_j$ | $C_j$ | $\xi_j$ |
| 1 | 1–52 | 30.43 | 1–87 | 30.42 | 1–79 | 29.99 |
| 2 | 53–93 | 30.37 | 88–123 | 30.18 | 80–130 | 30.45 |
| 3 | 94–128 | 30.38 | 124–151 | 30.33 | 131–162 | 29.57 |
| 4 | 129–159 | 30.33 | 152–180 | 29.89 | 163–185 | 30.58 |
| 5 | 160–287 | 30.15 | 181–205 | 29.96 | 186–206 | 29.46 |
| 6 | 188–213 | 30.35 | 206–227 | 29.57 | 207–226 | 29.43 |
| 7 | 214–237 | 30.02 | 228–247 | 29.49 | 227–246 | 30.77 |
| 8 | 238–259 | 29.21 | 248–266 | 30.32 | 247–265 | 30.47 |
| 9 | 260–280 | 29.39 | 267–284 | 30.79 | 266–283 | 29.98 |
| 10 | 281–300 | 29.36 | 285–300 | 29.06 | 284–300 | 29.31 |

**Table 3. Simulated means and variances of the Chi-squared test statistics.**

|  | LRS | | ERS | | BTS | | PTS | | SRS | | RRTS | |
|---|---|---|---|---|---|---|---|---|---|---|---|---|
|  | RWS | SUS | RWS | SUS | RWS | SUS | RWS | SUS | RWS | SUS | RWS | SUS |
| $\hat{\mu}$ | 8.68 | 0.0634 | 8.92 | 0.0542 | 8.71 | 0.0637 | 9.39 | 0.0647 | 9.35 | 0.0561 | 8.98 | 0.0492 |
| $\hat{\sigma}^2$ | 18.31 | 0.0003 | 17.94 | 0.0008 | 15.93 | 0.0002 | 20.48 | 0.0001 | 19.99 | 0.0003 | 17.92 | 0.0001 |

to assign selection probabilities to all the individuals and then one of two sampling schemes, i.e. $R$ is utilized to obtain instances of $o_i$, $O_j$ and $\chi^{S,R}$, respectively. By the succession of $(\chi_k^{S,R})_{1 \leq k \leq s}$, the sample mean and variance can be computed as:

$$\hat{\mu}^{(S,R)} = \frac{1}{s}\sum_{k=1}^{s}\chi_k^{S,R}, \tag{9}$$

$$\hat{\sigma}^{2(S,R)} = \frac{1}{s-1}\sum_{k=1}^{s}(\chi_k^{S,R} - \hat{e}^{(S,R)})^2. \tag{10}$$

For 99% confidence interval, it is compared with theoretical $\chi_{c-1}^2$ distribution. The sample mean and variance of chi-square should be close to $c - 1 = 9$, $2(c - 1) = 18$, respectively, and their estimates of $\hat{\mu}$ and $\hat{\sigma}^2$ are provided in Table 3. The SUS results are also reported in this table, which are shown its sampling accuracy as well. The average accuracy of the sampling method with all competing selection schemes is observed from these empirical results.

## 4.4 Empirical distribution function analysis

In this section, the empirical distribution function (EDF) is compared with theoretical chi-square distribution $\chi_{c-1}^2$ of roulette wheel sampling and it is given as:

$$EDF_{(t)}^{S,R} := \frac{1}{s}\{1 \leq k \leq s : \chi_k^{S,R} \leq t\}, \quad t \in [0, \infty). \tag{13}$$

In Fig 1, the behaviors of EDF (dashed line), for various selection operators for a population size $K = 300$ with a similar number of tests are reported. The selection operators are being compared with the theoretical $\chi_{c-1}^2$ distribution (dark thick line) using a 99% confidence band under the hypothesis of RWS (dashed thin double line). Here, the range on the x-axis values is $t \in [0, 18]$, where we expect the value of $\chi_k^{S,R}$, i.e. $E[\chi_k^{S,R}] = c - 1 = 9$. The RWS provides the empirical distribution function that is insignificant from the theoretical $\chi_{c-1}^2$ distribution by $\hat{e}$ and $\hat{\sigma}^2$ statistics. For sampling accuracy compared, the EDF of the proposed selection operator confirms a high sampling accuracy which is also proved in the statistics of Table 3.

## 5 Global performance

### 5.1 The traveling salesman problem

The most Illustrious benchmark, noteworthy and historic hard combinatorial optimization problem is the traveling salesman problem (TSP). In this problem, someone wants to find out the shortest Hamiltonian tour to starts his/her tour from a city and go to all other cities once and come back to the initial city. The first one, who documented this problem was Euler in 1759, see, for example, Larranaga et al. [26]. It is the most fundamental problem and has many applications in engineering, discrete mathematics, operations research, graph theory and computer science, etc.

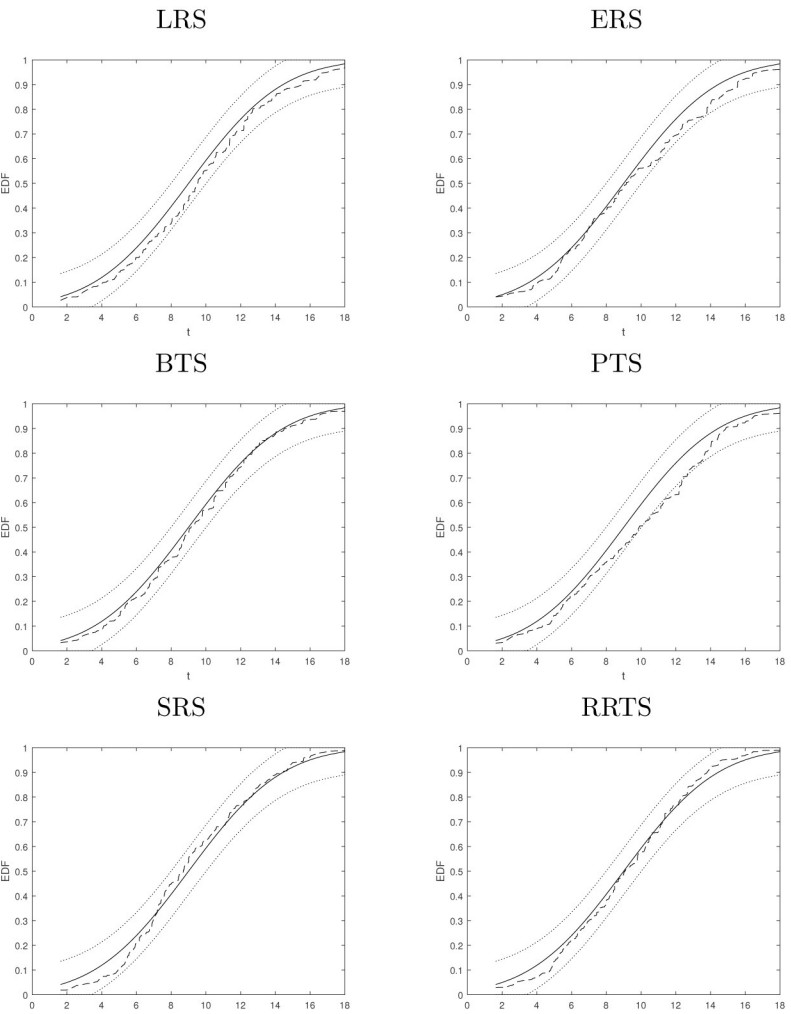

**Fig 1. Comparison of roulete wheel sampling, based on probabilities.**

Let $n$ cities with a distance (cost) matrix, $C = [c_{ij}]_{n \times n}$ is searched for a permutation $\lambda$: $\{0, \ldots, n-1\} \rightarrow \{0, \ldots, n-1\}$, where $c_{ij}$ is the distance between city $i$ and city $j$ and it minimizes the traveled distance $f(\lambda, C)$ as follows:

$$f(\lambda, C) = \sum_{i=0}^{n-1} d(c_{\lambda(i)}, c_{\lambda(i+1)}) + d(c_{\lambda(n)}, c_{\lambda(1)}), \tag{12}$$

where $\lambda(i)$ is the location of city $i$ in each tour, $d(c_i, c_j)$ is the distance from a city $i$ to another city $j$, whereas $(x_i, x_j)$ be a specified position of each city in a tour in the plane, and the Euclidean distances of the distance matrix $C$ between the city $i$ and city $j$ is extracted in the following way:

$$c_{ij} = \sqrt{(x_i - x_j)^2 + (y_i - y_j)^2}. \tag{13}$$

TSP is easy to understand but very difficult to solve, e.g. with '100' cities, there are $10^{155}$ possibilities to find the tour. This is the main reason to declare it as a non-deterministic polynomial

**Table 4. The benchmark problems.**

| Problem name | No. of cities | Optimal tour length |
|---|---|---|
| BERLIN52 | 52 | 7542 |
| PR144 | 144 | 58537 |
| KROB150 | 150 | 26130 |
| RAT195 | 195 | 2323 |
| KROA200 | 200 | 29368 |

(NP-hard) problem, see, for example, Hussain and Muhammad [13] and Hussain et al. [27]. Hence, this type of problem is not possible to solve using traditional optimization algorithms, e.g. gradient-based methods. To attain the optimal or close to optimal solution within an adequate amount of time, the heuristic algorithms are better choices to manage the NP-hard problems, see, Hussain and Muhammad [13], Huang et al. [28], Ruiz et al. [29]. The GA has also been applied for the solution of this problem in different ways, see, for example, Larranaga et al. [26], Hussain et al. [27], Hariyadi et al. [30], Alzyadat et al. [31], Dong and Cai [32]. Some test problems are taken from the library of traveling salesman problem (TSPLIB) for the global performance of the newly devised selection operator with respect to existing ones and reported in Table 4.

## 5.2 The state-of-the-art settings

For the simulation study, we used Windows PC with Intel i3 processor with 8 GB RAM, MATLAB R2017a software. Moreover, the two stopping criteria, i.e. not improvement found in 300 successive generations and the maximum number of generations (i.e. 5000) are used. The order crossover (OX) along with a well-known exchange mutation (EM) operator are used in this study. Table 5 is provided for further information about desired parameters.

## 5.3 Simulation results and discussion

In the above sections, the relative characteristics of the proposed operator with its competing selection methods with respect to sampling accuracy and population diversity have been determined. In this section, we test the performance of RRTS with other schemes by applying it to TSP. The results of six competing selection schemes with the most popular genetic operators, i.e. order crossover (OX) and exchange mutation (EM) are provided in Table 6, where all the tests are repeated thirty times. On the basis of average, standard deviation (S.D) and relative efficiency (R.E), these computational results are compared. Since TSP is a minimization

**Table 5. Parametric configuration for GA.**

| Parameter | Setting |
|---|---|
| Representation | Permutation |
| Population size | 150 |
| Crossover criteria | OX |
| Crossover rate | 80% |
| Mutation method | EM |
| Mutation rate | 5% |
| Maximum generation | 5000 |
| Number of trails | 30 |
| Replacement in GA | Steady-state GA |

problem, we observed an improved performance, based on 5000 simulations, by the proposed operator from among all six competing selection operators. From these results, we can confirm that RRTS outperforms the others.

The results listed in Table 6 demonstrate that the average tour length of the newly proposed selection operator (RRTS) is comparatively smaller than all other considered selection operators, with fewer S.D under all TSP instances. Hence, the empirical results of the simulation study prove that the average tour length under all considered TSP problems are not significantly divergent to the theoretical optimal tour length. This clearly shows a better control over selection pressure and population diversity. An excellent balance between exploration and exploitation is achieved by the RRTS, through maintaining an optimal convergence time, as compared to other operators.As a result, novel selection strategy outperformed other operators in terms of robustness, stability, and efficacy in solving complicated optimization problems. We also noted that, after altering and optimizing the parameters, the effectiveness of the simulation process is dependent on a wide range of parameters and measurement results. Based on this research, we suggest that the proposed operator may be used as a better alternative to get global optima or near to optimum results with minimal increase in complexity.

**Table 6. Results of various selection methods with respect to OX (crossover) and EM (mutation) operators.**

| Selection Method | Statistics | Problem | | | | |
|---|---|---|---|---|---|---|
| | | BERLIN52 | PR144 | KROB150 | RAT195 | KROA200 |
| FPS | Average | 7992 | 61856 | 28481 | 2497 | 31125 |
| | S.D | 293 | 1777 | 1007 | 99 | 961 |
| | R.E | 5.97 | 5.67 | 9.00 | 7.49 | 5.98 |
| | Ave. time (ms) | 12.3 | 19.1 | 21.5 | 27.6 | 31.3 |
| LRS | Average | 8071 | 61597 | 27803 | 2425 | 30503 |
| | S.D | 369 | 1543 | 913 | 113 | 914 |
| | R.E | 7.01 | 5.23 | 6.40 | 4.39 | 3.86 |
| | Ave. time (ms) | 13.7 | 21.3 | 23.2 | 28.8 | 34.5 |
| ERS | Average | 8458 | 63766 | 29581 | 2518 | 31813 |
| | S.D | 441 | 1721 | 883 | 138 | 747 |
| | R.E | 12.15 | 8.93 | 13.21 | 8.39 | 8.33 |
| | Ave. time (ms) | 12.6 | 18.8 | 21.4 | 26.5 | 31.8 |
| BTS | Average | 7976 | 60702 | 27562 | 2453 | 29920 |
| | S.D | 341 | 1239 | 958 | 113 | 865 |
| | R.E | 5.75 | 3.70 | 5.48 | 5.60 | 1.88 |
| | Ave. time (ms) | 11.9 | 18.4 | 20.8 | 26.4 | 29.9 |
| PTS | Average | 8021 | 60797 | 28201 | 2415 | 30873 |
| | S.D | 451 | 1001 | 937 | 115 | 882 |
| | R.E | 6.35 | 3.86 | 7.93 | 3.96 | 5.12 |
| | Ave. time (ms) | 12.4 | 17.6 | 20.8 | 25.7 | 30.4 |
| SRS | Average | 7998 | 60701 | 27603 | 2422 | 29959 |
| | S.D | 309 | 1011 | 857 | 98 | 842 |
| | R.E | 6.05 | 3.70 | 5.64 | 4.26 | 2.01 |
| | Ave. time (ms) | 12.6 | 18.5 | 19.5 | 26.9 | 30.3 |
| RRTS | Average | 7957 | 60627 | 27548 | 2400 | 29941 |
| | S.D | 297 | 1014 | 883 | 105 | 828 |
| | R.E | 5.50 | 3.57 | 5.43 | 3.31 | 1.95 |
| | Ave. time (ms) | 12.9 | 18.7 | 19.9 | 26.5 | 30.8 |

Moreover, researchers may feel comfortable to apply it to any problems related to evolutionary algorithms.

## 6 Conclusions

For every optimization algorithm, the main desire is to balance between two extremes, i.e. exploration and exploitation. This article presents a new round-robin based tournament selection operator for GAs, which is suggested a fine balance between exploitation and exploration. The individuals are sorted with respect to their fitness measures and then the whole population is divided into two equal and non-overlapping groups, i.e. $A$ and $A^c$. To determine the sampling accuracy, we employ $\chi^2$ test to confirm a close match between the expected and observed number of offspring (insignificant difference). A simulation study is performed to evaluate the performance of the newly devised selection operator along with some conventional operators. Based on this research, we suggest that the proposed operator might be used as a better alternative to get global optima or near to optimum results. Moreover, researchers might be apply it for any problems related to evolutionary algorithms.

## Acknowledgments

The author thanks the honorable editor and reviewers for their constructive comments and suggestions which helped the author to improve this paper.

## Author Contributions

**Conceptualization:** Abid Hussain.

**Data curation:** Abid Hussain, Ehtasham ul Haq.

**Formal analysis:** Abid Hussain, Salma Riaz, Ehtasham ul Haq.

**Funding acquisition:** Salma Riaz, Muhammad Sohail Amjad, Ehtasham ul Haq.

**Investigation:** Muhammad Sohail Amjad.

**Methodology:** Abid Hussain.

**Resources:** Muhammad Sohail Amjad.

**Software:** Abid Hussain, Salma Riaz, Ehtasham ul Haq.

**Supervision:** Abid Hussain.

**Validation:** Abid Hussain, Muhammad Sohail Amjad.

**Writing – original draft:** Abid Hussain.

**Writing – review & editing:** Salma Riaz, Muhammad Sohail Amjad.

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
