## [Decision Letter · Decision Letter 0]

17 May 2022

PONE-D-22-05986Genetic algorithm with a new round-robin based tournament selection: statistical properties analysisPLOS ONE

Dear Dr. Hussain,

Thank you for submitting your manuscript to PLOS ONE. After careful consideration, we feel that it has merit but does not fully meet PLOS ONE’s publication criteria as it currently stands. Therefore, we invite you to submit a revised version of the manuscript that addresses the points raised during the review process.

ACADEMIC EDITOR: Please consider all the comments from the reviewers and proof read the revised manuscript carefully before resubmitting it.

We look forward to receiving your revised manuscript.

Kind regards,

Qichun Zhang, PhD

Academic Editor

PLOS ONE

Journal Requirements:

2. Please update your submission to use the PLOS LaTeX template. The template and more information on our requirements for LaTeX submissions can be found at http://journals.plos.org/plosone/s/latex

Additional Editor Comments:

Based on the review comments, a major revision is needed for the current manuscript. Basically, the comparative study is important for the revised version and the writing needs to be improved as well. In addition, the reference seems not up-to-date, please update the list with the latest contributions.

Reviewers' comments:

Reviewer's Responses to Questions

**Comments to the Author**

1. Is the manuscript technically sound, and do the data support the conclusions?

Reviewer #1: Yes

Reviewer #2: Partly

Reviewer #3: Yes

2. Has the statistical analysis been performed appropriately and rigorously? 

Reviewer #1: Yes

Reviewer #2: No

Reviewer #3: Yes

3. Have the authors made all data underlying the findings in their manuscript fully available?

Reviewer #1: Yes

Reviewer #2: Yes

Reviewer #3: No

4. Is the manuscript presented in an intelligible fashion and written in standard English?

Reviewer #1: Yes

Reviewer #2: No

Reviewer #3: Yes

5. Review Comments to the Author

Reviewer #1: Subtopic 4.1 - line 1 - the citation for author "Holland" incomplete (i.e., missing year)

Subtopic 4.2 - add short-form (SUS) after the title

Subtopic 4.2 - line 1 - the citation for author "Baker" incomplete (i.e., missing year) and revise the whole sentence.

Please refer to the attachment for correction on minor grammatical errors.

Reviewer #2: The motivation is clear. I have the following concerns:

1. I suggest more sufficient interpretation to the statistical analysis.

2. I suggest sufficient comparison between literature and the manuscript, especially in results.

Reviewer #3: This paper proposes a round-robin based tournament selection operator for the genetic algorithms.

1. The contribution is missing. The authors should add a paragraph in the introduction section to clearly describe the contribution of the work.

2. The literature review is not sufficient. Most cited references are published before 2016. The latest literature year is the previous work of the author. The authors need to track the latest progress.

3. Table 6 only analyzes the statistics. But the computational costs are also important. The authors need to compare the minutes of run-time of the proposed method to that of the compared methods.

6. PLOS authors have the option to publish the peer review history of their article (what does this mean?). If published, this will include your full peer review and any attached files.

Reviewer #1: No

Reviewer #2: No

Reviewer #3: **Yes: **Jiaxin Cai

---

## [Author Response · Author response to Decision Letter 0]

1 Jul 2022

A separate file of responses has been uploaded in attach files section.

---

## [Decision Letter · Decision Letter 1]

30 Aug 2022

Genetic algorithm with a new round-robin based tournament selection: statistical properties analysis

PONE-D-22-05986R1

Dear Dr. Hussain,

We’re pleased to inform you that your manuscript has been judged scientifically suitable for publication and will be formally accepted for publication once it meets all outstanding technical requirements.

Kind regards,

Qichun Zhang, PhD

Academic Editor

PLOS ONE

Additional Editor Comments :

The reviewers satisfied the revision with no further comment. Personally, I went through the revision and the response. I believe that the revised version has been improved strongly in terms of quality and readability. The contribution and novelty have been highlighted. Based on the results, I would like to support this submission and recommend accepting this manuscript.

Reviewers' comments:

Reviewer's Responses to Questions

**Comments to the Author**

1. If the authors have adequately addressed your comments raised in a previous round of review and you feel that this manuscript is now acceptable for publication, you may indicate that here to bypass the “Comments to the Author” section, enter your conflict of interest statement in the “Confidential to Editor” section, and submit your "Accept" recommendation.

Reviewer #2: All comments have been addressed

2. Is the manuscript technically sound, and do the data support the conclusions?

Reviewer #2: Yes

3. Has the statistical analysis been performed appropriately and rigorously? 

Reviewer #2: Yes

4. Have the authors made all data underlying the findings in their manuscript fully available?

Reviewer #2: Yes

5. Is the manuscript presented in an intelligible fashion and written in standard English?

Reviewer #2: Yes

6. Review Comments to the Author

Reviewer #2: All my previous concerns were taken into account and suitably responded, including sufficient interpretation to the statistical analysis et al.

7. PLOS authors have the option to publish the peer review history of their article (what does this mean?). If published, this will include your full peer review and any attached files.

Reviewer #2: No

---

## [Editor Report · Acceptance letter]

31 Aug 2022

PONE-D-22-05986R1 

Genetic algorithm with a new round-robin based tournament selection: statistical properties analysis 

Dear Dr. Hussain:

I'm pleased to inform you that your manuscript has been deemed suitable for publication in PLOS ONE. Congratulations! Your manuscript is now with our production department. 

Kind regards, 

on behalf of

Dr. Qichun Zhang 

Academic Editor

PLOS ONE